Citation: *Molecular Systems Biology* 9:656
www.molecularsystemsbiology.com

# Increasing population growth by asymmetric segregation of a limiting resource during cell division

Nurit Avraham, Ilya Soifer, Miri Carmi and Naama Barkai*

Department of Molecular Genetics, Weizmann Institute of Science, Rehovot, Israel
* Corresponding author. Department of Molecular Genetics, Weizmann Institute of Science, Rehovot 76100, Israel. Tel: + 972 8 934 4429; Fax: + 972 8 934 4108; E-mail: naama.barkai@weizmann.ac.il

We report that when budding yeast are transferred to low-metal environment, they adopt a proliferation pattern in which division is restricted to the subpopulation of mother cells which were born in rich conditions, before the shift. Mother cells continue to divide multiple times following the shift, generating at each division a single daughter cell, which arrests in G1. The transition to a mother-restricted proliferation pattern is characterized by asymmetric segregation of the vacuole to the mother cell and requires the transcription repressor Whi5. Notably, while deletion of *WHI5* alleviates daughter cell division arrest in low-zinc conditions, it results in a lower final population size, as cell division rate becomes progressively slower. Our data suggest a new stress-response strategy, in which the dilution of a limiting cellular resource is prevented by maintaining it within a subset of dividing cells, thereby increasing population growth.
*Molecular Systems Biology* (2013) **9,** 656; published online 16 April 2013; doi:10.1038/msb.2013.13
*Subject Categories:* cellular metabolism; differentiation & death
*Keywords:* budding yeast; nutrients limitation; phenotypic diversity; zinc

## Introduction

Rapid adaptation to fluctuations in nutrient availability is critical to all cells and in particular to microorganisms that live in a constantly changing environment. Maintaining homeostasis of transition metals (e.g., zinc, iron or copper) is particularly critical (Eide, 2001; Rutherford and Bird, 2004). The unique chemical and physical properties of transition metals provide essential biochemical activities and structural motifs to a multitude of proteins, but also render them toxic to the cells when present in excess. In the budding yeast, e.g., zinc is an essential catalytic component (reviewed in Eide, 2003; Eide, 2009) of over 300 enzymes and is important for the correct folding of structural motifs, such as zinc fingers. Toxicity to excess of zinc likely results from its binding to inappropriate sites in proteins or cofactors, which interfere with their normal function.

Cells employ regulatory mechanisms that sense when specific nutrients are depleted, and adjust gene expression, protein activities or metabolic preferences to adapt to this limitation. For example, the Zap1 transcription factor is activated upon zinc depletion, resulting in the upregulation of genes involved in high-affinity zinc transport (Eide, 2006, 2009) or zinc storage in the vacuole (Simm *et al*, 2007). Similarly, under conditions of iron deficiency, the partially redundant Aft1 and Aft2 transcription factors are activated and induce genes involved in iron homeostasis (Kosman, 2003).

Maintaining homeostasis requires adaptation that is specific to the nutrient being depleted. In addition, cells adapt by modulating their division cycle or by initiating differentiation processes. For example, budding yeast starved for nitrogen or carbon become filamentous, a growth mode that may facilitate food foraging (Gimeno *et al*, 1992). Additional differentiation path triggered by nitrogen starvation is meiosis and sporulation (reviewed in Neiman, 2011), which ensures a protected environment for long-term survival.

Limitations of one nutrient may indicate a more general deterioration of the external environment and perhaps the onset of an uncertain period. Theoretical studies suggest that when conditions fluctuate in an unpredictable way, the overall population fitness can be increased if cells diversify into subpopulations that are individually optimized for a subset of possible conditions (Thattai and van Oudenaarden, 2004; Kussell and Leibler, 2005). Stress-induced population splitting was observed in bacteria, caused by either asymmetric cell division or by stochastic processes (Balaban *et al*, 2004; Suel *et al*, 2006; Acar *et al*, 2008; William and Denison, 2010).

Cell division is inherently asymmetric in budding yeast, and this asymmetry is amplified under nutrient limitations (Lord and Wheals, 1981). As division is by budding, the identities of the mother and daughter cells are well defined. Mothers are larger and display a short G1. In contrast, daughters are born small and spend a longer time in G1, during which they grow to about the size of their mothers. Nutrient limitation generally prolongs the G1 phase and increases the inherent asymmetry in cell size and in cell-cycle duration between mothers and daughters. Indirect evidence suggests that upon glucose

depletion, daughter cells become quiescent before entering stationary phase (Allen *et al*, 2006; Werner-Washburne *et al*, 1993; Petti *et al*, 2011). However, it is not known whether this population splitting into mothers and daughters provides some advantage for population growth or survival.

We asked whether cell-cycle regulation has a role in the adaptation of budding yeast to a low-metal environment. To this end, we used automated video microscopy to follow individual cells shifted from a rich medium to an environment with low levels of one of the transition metals, zinc, iron or copper. We report that cells adopt a new proliferation pattern in which division is restricted to a subpopulation of mother cells. The vacuole, a central site for nutrient storage, is retained by the mother cell upon division, in contrast to its symmetric segregation between mother and daughter when nutrients are abundant. This division pattern may implement a risk-spreading strategy, as mothers and daughters display different survival capabilities when exposed to different types of stresses. We propose, however, that the main role of this asymmetric division is to prevent the dilution of a limiting resource, thereby maximizing the overall population growth in low-nutrient environment.

## Results

### Splitting of budding yeast population into dividing mother cells and non-dividing daughter cells

We monitored growth and division of yeast cells while shifting them from a rich medium (synthetic-complete) to a medium

containing low levels of zinc. Our automated microscopy setup was coupled to a microfluidic flow cell that enables growing cells in a planar layer for several days, while continuously providing them with fresh medium (Charvin *et al*, 2008, Supplementary Figure 1a). When subject to non-limiting conditions, the number of cells increased exponentially with an average doubling time of 103 min (Figure 1A). Surprisingly, upon transfer to a low-zinc medium (LZM + 10 µM $Zn^{2+}$), the increase in cell number became linear in time (Figure 1B, Supplementary Figure 2a). The same pattern of linear growth was observed also when cells were shifted to a medium with low concentration of iron (Figure 1C, Supplementary Figure 2c) or copper (Figure 1D, Supplementary Figure 2e), two additional transition metals. Cells shifted to low-phosphate medium maintained an exponential growth (Figure 1E).

Linear growth may result from an increase in doubling time between subsequent divisions. Alternatively, it may indicate that division is restricted to a fixed number of cells only. To identify the dividing cells, we fluorescently labeled the bud neck and the nucleus (Bean *et al*, 2006) (using the protein fusions Cdc10–GFP and Acs2–mCherry, respectively), and wrote an image analysis software that automatically segments cells and constructs their lineage tree. Monitoring cells upon transfer to low zinc revealed that division is restricted to mother cells that were born before the transfer and their very early progenies (Figures 2A–C, Supplementary Figure 2b, Supplementary Movie 1). Daughter cells born in low zinc (more than 2 h after the transition) did not divide during our experiment (Figure 2C), while their mothers continued to

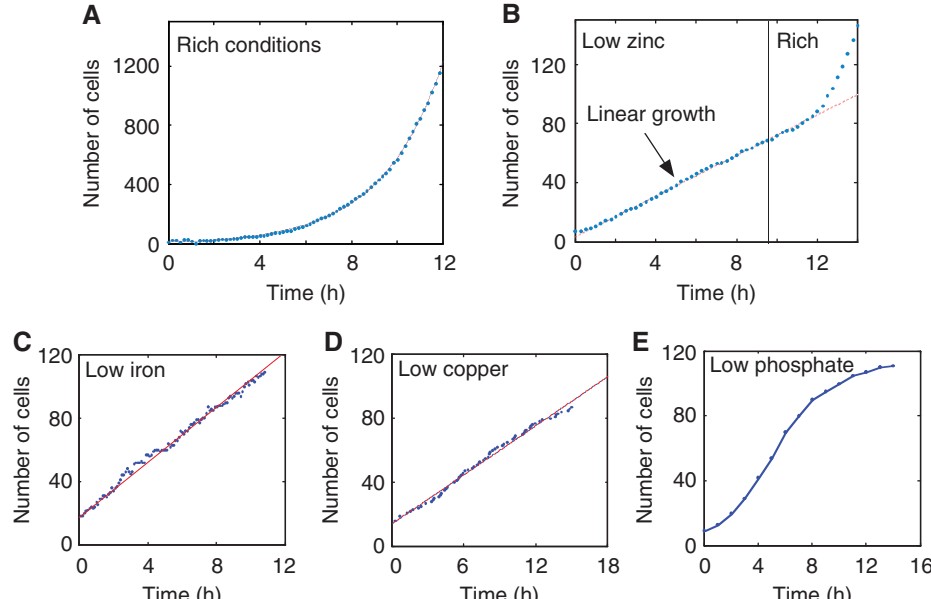

**Figure 1** Linear growth in low-metal environment. (**A**) Exponential growth observed in non-limiting conditions (rich). Cells were grown in rich medium and were counted using automatic image analysis software. Shown is the number of cells in a field of view. The red dashed line is an exponential fit ($r^2 = 0.9994$). Doubling time of 103 min was extracted from the fit. (**B**) Linear growth observed in LZM. Cells were grown in low-zinc conditions (LZM + 10 µM $Zn^{2+}$) and counted as above. Zinc was replenished at $t = 10$ h, following which the cells resumed exponential growth. The dashed red line is a linear fit to the linear part of the growth curve ($r^2 = 0.9985$), see also Supplementary Figure 1b. Linear behavior was observed also in batch cultures (see Supplementary Figure 1b). We performed the same experiment where for the rich medium we used LZM + 300 µM $Zn^{2+}$ instead of SC (see Supplementary Figures 2a and b). (**C**) Linear growth observed in low-iron conditions. Cells were grown in low-iron conditions (Methods) and counted as above over three different fields of view. (**D**) Linear growth observed in low-copper conditions. Same as (C) for low-copper conditions (Methods). (**E**) No linear growth was observed in low phosphate conditions. Same as (C) for low-phosphate conditions (Methods).

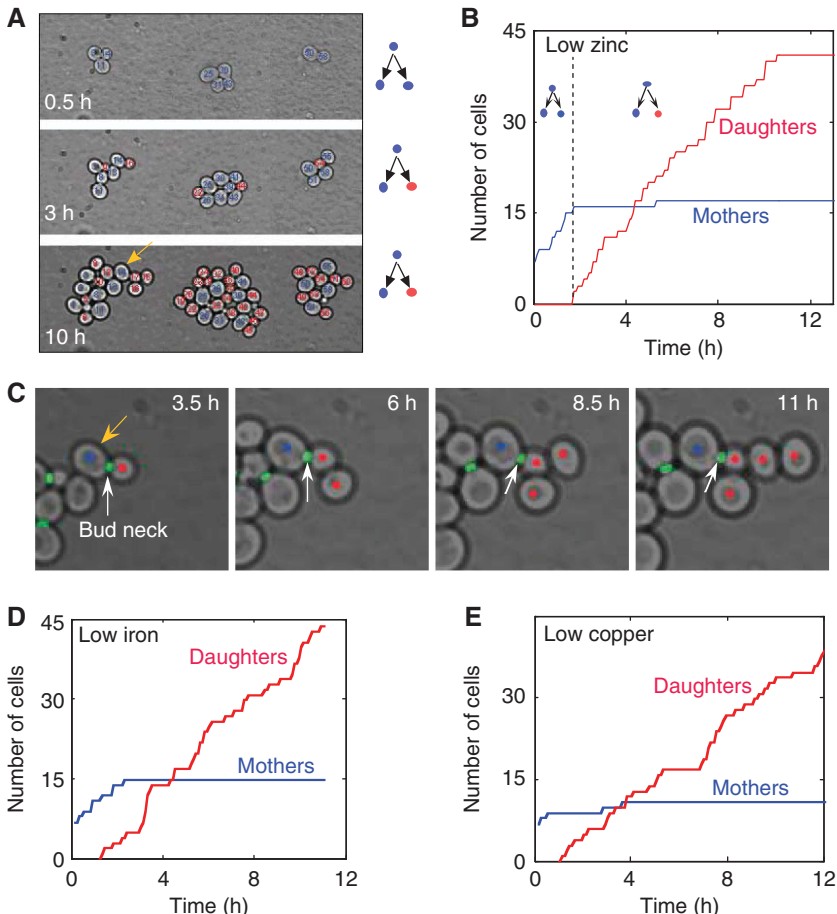

**Figure 2** Splitting of budding yeast population into dividing mother cells and non-dividing daughter cells in low-zinc conditions. (**A**, **B**) Population splitting into dividing mother cells and non-dividing daughter cells. Cells were shifted to low zinc (LZM + 10 μM Zn²⁺) at *t* = 0, following pre-adaptation in an intermediate zinc concentration (see Supplementary experimental procedure for details). Snapshots of a typical colony are shown in (A) with cells that divided at least once (mothers) colored blue, and those that have not (daughters) colored red (see Supplementary Movie 1). Quantification of the dynamics, showing the increase in the number of mother and daughter cells in multiple colonies, is shown in (B). Note that mothers divide asymmetrically for 4–5 generations, at a doubling time of ∼150 min. When shifted directly from rich conditions to low zinc (Supplementary Figure 1b), they divide asymmetrically for about 8 times at a doubling time of ∼120 min. (**C**) Daughter cells born in low zinc do not divide. A series of snapshots showing a mother cell (blue, marked by a yellow arrow in A), giving rise to four non-dividing daughter cells (red). Note the bud-neck marker that indicates division of the mother-only. (**D**) Splitting into mothers and daughters in low-iron conditions. Shown is the increase in the number of mothers (red) and daughters (blue) in multiple colonies summarized over two fields of view out of the three presented in (Figure 1C). (**E**) Linear growth observed in low-copper conditions. Same as (D) for low-copper conditions (Methods).

divide in low zinc up to eight times, and maintained a practically constant doubling time of 120–150 min before arresting. Once again, the same division pattern was observed also in cells transferred to a low iron (Figure 2D, Supplementary Figure 2d) or low copper (Figure 2E, Supplementary Figure 2f) media. Also here, the division was restricted to mother cells born before the transfer.

To examine whether the non-dividing daughter cells lose the capacity for protein expression, we followed the induction of the affinity zinc transporter Zrt1, which is upregulated upon zinc depletion. A fluorescent GFP reporter driven by the *ZRT1* promoter was strongly induced in both mother and daughter cells, and this induction was indistinguishable between the dividing and non-dividing cells (see Supplementary Figure 1d). We also examined for differential cell viability, by shifting the cells back to a rich media (SC). Both mothers and daughters resumed normal division upon this transfer (Figure 1B, *t* > 12 h).

## Whi5p is required for the division arrest of daughter cells

The non-dividing daughter cells were smaller than the mother cells by ∼35% (Figure 3A). This size difference was maintained when accounting for their age difference (Figure 3A inset). We therefore hypothesized that the transcription repressor Whi5, which prolongs G1 in small cells, prevents daughters from dividing in these conditions. Under normal conditions, cells deleted of *WHI5* grow at a normal rate (Figure 3E), but are ∼30% smaller than wild type (Costanzo *et al*, 2004; de Bruin *et al*, 2004). When transferred to low zinc, we found that deletion of *WHI5* eliminated the population splitting: all cells were now of the same average size (Figure 3B), and most cells divided, even daughter cells born in low zinc (Figure 3C).

Considering the fact that wild-type cells restricted proliferation to mother cells only, while in *whi5* most cells divided, we

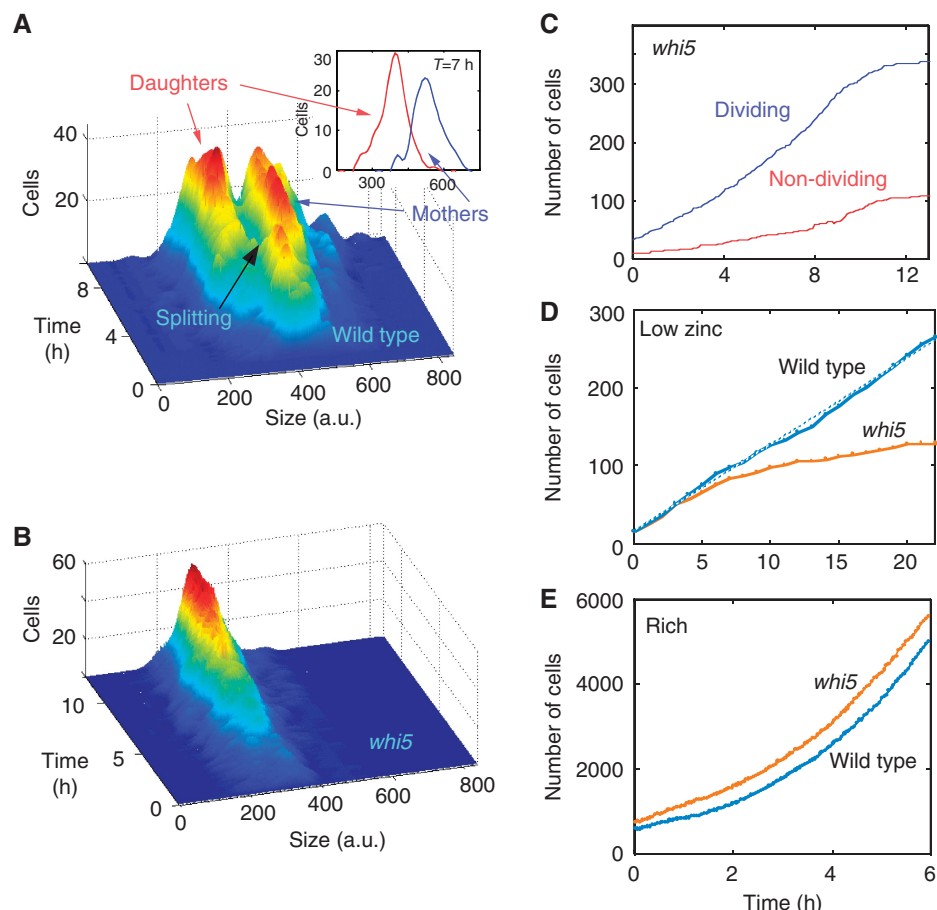

**Figure 3** No population splitting in *whi5*. (**A**) Mothers and daughters differ in cell size. The distributions of cell sizes of wild-type mothers and daughters at different times following the transfer to LZM. The inset shows the distribution at $t = 7$ h after accounting for age differences. (To make sure the size difference is not owing to the fact that mothers are older than the daughters, we treated the single-cell growth curves as if all cells were born at the same time.) (**B**) Deletion of WHI5 eliminates the splitting into subpopulations. Same as (A), for *whi5* cells. (**C**) Deletion of WHI5 enables division of daughter cells. Increase in the number of dividing (blue) and non-dividing (red) cells in multiple colonies of *whi5* cells. Note that in contrast to the wild-type where the number of dividing cells (Figure 2b, blue) stops increasing about 2 h after the shift, in *whi5* the number of dividing cells continues to grow long time after the shift. Thus, even daughter cells that were born in low zinc divide. (**D**, **E**) Proliferation of wild-type and *whi5* cells. Growth curves of wild-type and *whi5* cells grown together in the same flow cell in low zinc (**D**) and rich (**E**) media. See also Supplementary Figure 3.

expected *whi5* cells to outcompete the wild-type cells in low zinc. This, however, was not the case: in low zinc, wild-type cells generated a larger population than *whi5* cells (Figure 3D). This difference in population growth was explained when we measured the duration of the cell division cycle: wild-type mothers maintained essentially the same division rate for up to eight cycles, while doubling time in *whi5* increased with each subsequent division leading to a lower overall proliferation in low zinc.

## Vacuole size correlates with the capacity to divide

Further comparison of the wild-type and *WHI5*-deleted cells revealed that the capacity to divide correlates with the cell vacuole size. At low zinc, mother cells maintained most of the vacuole so that daughter cells received extremely small vacuole (Figures 4A and B). In contrast, *WHI5*-deleted cells split the vacuole proportionally between mothers and daughters (Figures 4C and D), similarly to wild-type cells growing in

rich medium (Catlett and Weisman, 2000) (Figure 4E). This correlation between vacuole size and cell capacity to divide was strengthened by the observation that very early daughter cells that were still able to divide, had large vacuoles of size similar to that of their mothers (Figure 4A at $t < 3$ h). Furthermore, mutants impaired in vacuolar functions, in which both mothers and daughters had defective vacuoles, stopped dividing immediately when shifted to low zinc conditions (Figure 4F).

To further study the relationship between asymmetric distribution of the vacuole and the capacity to divide, we examined the growth and division in cells deleted of *VAC17*, which impairs the transport of the vacuole toward the bud, thereby disrupting its segregation to daughter cells (Tang *et al*, 2003). We asked whether this intrinsic asymmetry of vacuole segregation in *vac17* cells promotes mother-restricted proliferation even in conditions where wild-type cells are still capable of exponential growth. In rich conditions, daughter cells of the *vac17* genotype rapidly generated new vacuoles (Raymond *et al*, 1990) and did not show any growth defect

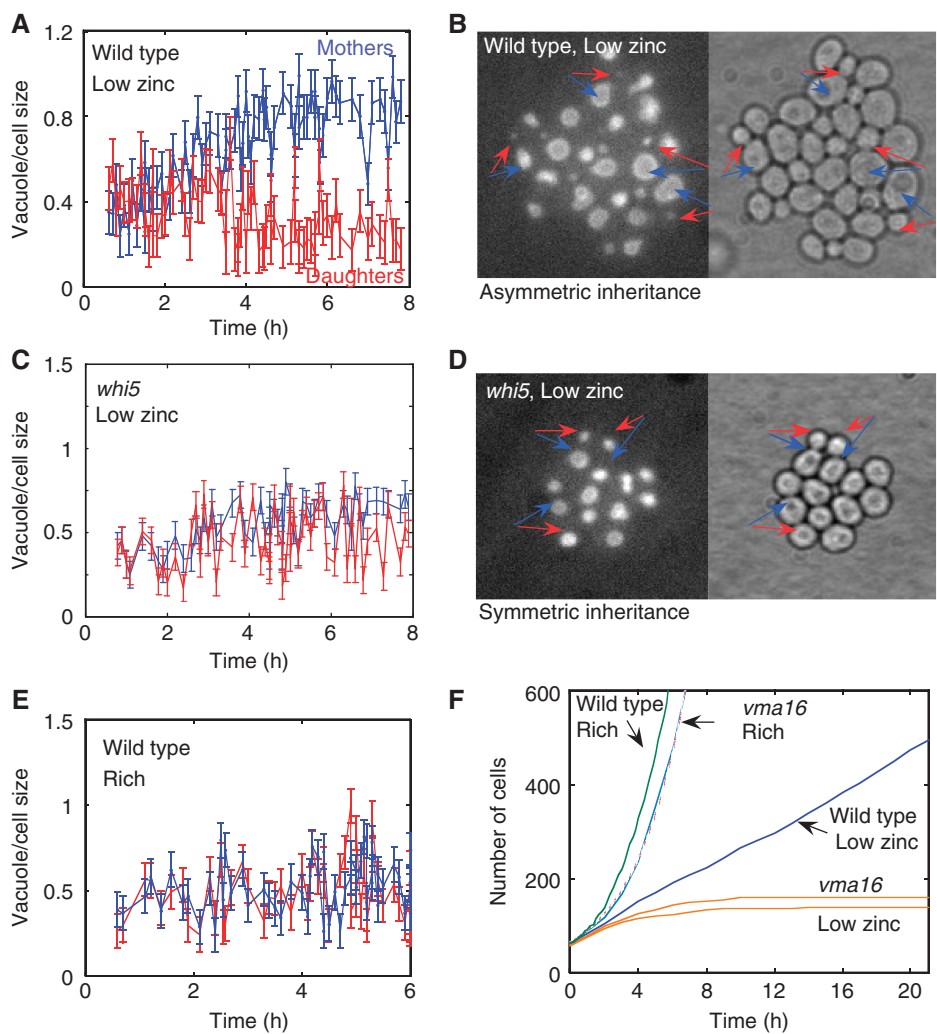

**Figure 4** Correlation between vacuole size and capacity to divide. (**A**, **B**) Mothers maintained most of the vacuole in low zinc. Shown is the fraction of vacuole that mothers (blue) and daughters (red) receive upon division, as extracted from fluorescent images (B) of cells containing vacuolar marker (Vph1-GFP). The arrows indicate mother–daughter pairs just after division. Note that early divisions that occurred before the splitting ($t < 3$ h) distribute the vacuole symmetrically between the mother and the daughter cells. In contrast, divisions that occurred at $t > 3$ h lead to asymmetric distribution, where mothers maintained a larger portion of the vacuole. (**C**, **D**) Symmetric vacuole inheritance in *whi5* cells in low zinc. Same as (A, B) for *whi5* cells. (**E**) Symmetric vacuole inheritance in exponentially growing cells. Same as (A) for wild type in rich conditions. (**F**) Mutant with defective vacuole (*vac17*) does not divide in low zinc. Shown are growth curves of wild type in rich (green) and low-zinc conditions (blue), *vma16* in rich (blue, with red dashed fitting curve) and low-zinc conditions (orange, two different experiments).

(Figure 5A). However, in semi-low conditions (LZM + 300 μM $Zn^{2+}$), where wild-type cells displayed a slower but still exponential growth (Figures 5B and C), *vac17* cells that distribute the vacuole asymmetrically (Figures 5D and E) displayed a linear growth (Figure 5B): daughters cells had small or no vacuoles, and did not divide (Figure 5F, Supplementary Figure 4 and Supplementary Movie 2). This is likely explained by the fact that in semi-low conditions, vacuole regeneration was significantly impaired and therefore daughters that did not receive vacuoles owing to the *VAC17* deletion could not generate new vacuoles.

The vacuoles serve as a central site for nutrient storage, and in particular store an excess of zinc (Supplementary Figure 5) and other transition metals (MacDiarmid *et al*, 2000; Simm *et al*, 2007). A compelling hypothesis is that those stored pools are required for enabling division upon transfer to low-nutrient conditions. To test this hypothesis, we examined cells

deleted of the two vacuolar transporters Cot1 and Zrc1 required for zinc storage. As expected, mother-cell growth rate was reduced in the *cot1-zrc1* cells, while daughter cell division arrest was maintained under low-zinc conditions (10 μM $Zn^{2+}$) (Figures 6A and B). Further, the *cot1-zrc1* cells underwent the transition to linear growth earlier, in conditions (50 μM $Zn^{2+}$) where wild-type cells still display an exponential proliferation (Figures 6C and D). We conclude that the vacuolar zinc pool has a role in enabling proliferation in low zinc.

## Differential stress sensitivity of the dividing versus non-dividing cells

In addition to serving as a site for nutrient storage, the yeast vacuoles protect cells against a variety of stresses. We

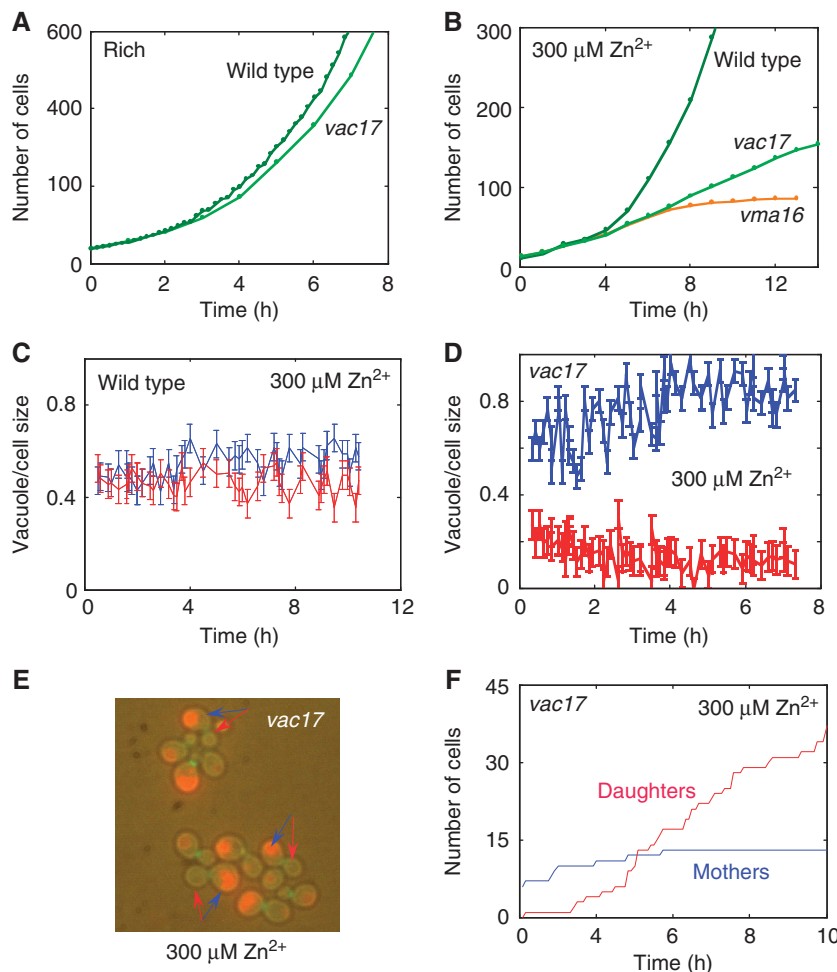

**Figure 5** Asymmetric vacuole inheritance promotes mother-only proliferation. (**A**) Exponential growth of wild type and *vac17* in rich conditions: in rich condition *vac17* (light green) grow exponentially similar to wild type (green). (**B**) *vac17* displays linear growth in semi-low conditions. In semi-low conditions (LZM + 300 μM Zn), wild-type cells grow exponentially while *vac17* display linear growth with an average doubling time of ∼180 min. Note that *vma16* where both mothers and daughters have defective vacuole decays rapidly. (**C**) Symmetric vacuole inheritance in wild type in semi-low conditions. Shown is the fraction of the vacuole mothers and daughters receive upon division (same as in Figure 4a). (**D**) Asymmetric vacuole inheritance in *vac17* in semi-low conditions. Same as (C) in *vac17*. (**E**) Fluorescent image of *vac17* cells containing vacuolar and budneck markers. The arrows indicate mother (blue)—daughter (red) pairs just after division. (**F**) Splitting into mothers and daughters in *vac17*. Shown is the increase in number of mothers (blue) and daughters (red) in *VAC17*-deleted cells.

therefore reasoned that mothers and daughters, being different in proliferation capacity and vacuolar content, may be differentially adapted to cope with additional stresses. To test this, we first considered vacuole-related protections, which are predicted to be better endured by mother cells. We subjected cells to a zinc shock (MacDiarmid *et al*, 2003), by transferring the cells back to a medium containing a very high zinc concentration (SC + 9 mM Zn$^{2+}$). The dividing mothers emerged from the initial arrest and resumed normal growth, while daughter cells recovered only partially and after a significant delay (Figure 7A). Similarly, mothers better resisted osmotic stress (Li *et al*, 2012) when exposed to NaCl, about 30% of the daughters, but only 8% of the mothers, imploded during the time course of experiment (Figure 7B, Supplementary Movies 3a and b).

While mother cells better resist stresses requiring vacuole protection (Li *et al*, 2012), daughter cells may better resist other stresses, in particular drugs that affect dividing cells.

Consistent with that, daughters better recovered from rapamycin, a drug which inhibits the TOR kinases regulating cell growth, proliferation and survival (Crespo and Hall, 2002). When exposed to the drug for a few hours, and then shifted back to rich conditions, both mothers and daughters attempted to divide, as indicated by the appearance of a bud neck ring (not shown). However, only daughter cells managed to complete the cell cycle, while mothers mostly imploded (Figures 7C and D, Supplementary Movie 3c).

## Discussion

In budding yeast, cell division is inherently asymmetric, generating mother and daughter cells that differ in their size and in the duration of the G1 phase. This asymmetry becomes more pronounced in non-optimal conditions that reduce cell growth rate and prolong G1. We now report that under

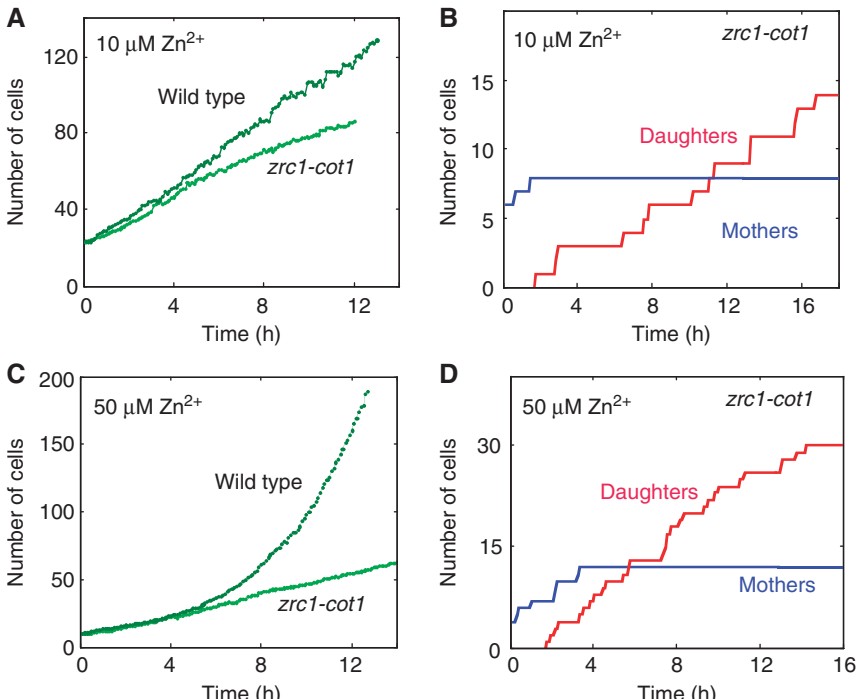

**Figure 6** The effect of the vacuolar zinc pool size on the transition timing. (**A**) Growth of wild type and *zrc1-cot1* in low zinc conditions: in low zinc (LZM + 10 μM $Zn^{2+}$), mother cell growth rate in *zrc1-cot1* was reduced. (**B**) Splitting into mothers and daughters in *zrc1-cot1* in low zinc (LZM + 10 μM $Zn^{2+}$). Shown is the increase in number of mothers (blue) and daughters (red) in *zrc1-cot1* cells. (**C**) Growth of wild type and *zrc1-cot1* in less severe conditions (LZM + 50 μM $Zn^{2+}$), wild-type cells grow exponentially while *trc1-cot1* cells display linear growth. (**D**) Splitting into mothers and daughters in *zrc1-cot1* in less severe conditions (LZM + 50 μM $Zn^{2+}$). Shown is the increase in number of mothers (blue) and daughters (red) in *zrc1-cot1* cells.

conditions of metal limitation, mother–daughter asymmetry is further extended, such that daughter cells arrest their division cycle completely, whereas mother cells continue to divide. We observe that when dividing in low zinc, mother cells maintain most of the vacuoles, in contrast to the more equal vacuole distribution between mother and daughters seen in rich media. Further, regeneration of vacuoles under those conditions is apparently slow, leaving daughter cells with very small vacuoles. These small vacuoles may explain daughter cell division arrest, perhaps by devoiding them from stored zinc pools.

The transition to linear population growth appears to be regulated, as the mother–daughter asymmetry is lost upon deletion of the cell-cycle regulator *WHI5*. What could be the possible benefit of restricting division to one set of progenitor cells? The phenotype of *WHI5*-deleted cells may be suggestive of such a role. In this mutant, both mothers and daughters divide, yet overall population growth is reduced relative to wild type owing to a progressive extension of their cell cycle. Vacuoles in this mutant are distributed proportionally between mother and daughters, which may result in their progressive dilution.

We propose that division is restricted to mother cells to prevent dilution of some essential resource. This resource is likely to be contained in the vacuole or, more specifically, the available zinc that is stored in the vacuole. A simple mathematical model suggests that under low-nutrient conditions, when resources are not easily replenished, such a strategy maximizes population growth. Assume, e.g., that cells

enter a poor environment where some limiting resource needed for division cannot be replenished. If the initial level of this limiting resource is $Z_0$ and cells divide symmetrically, the level of resource will decrease twofold at each division, approaching the limit $Z_C$ under which division is not possible in just $\log_2(Z_0/Z_C)$ cycles, leading to a maximal population growth by a factor $N_s = Z_0/Z_C$ cells. In contrast, if the progenitor cell maintains all the resource, division can proceed indefinitely. This simple model can easily be extended to perhaps a more realistic situation where an amount $\Delta Z$ of the limiting resource is lost at each division, in which case the maximal number of progenies generated by each asymmetrically dividing progenitor will be $N_{as} = (Z_0 - Z_c)/\Delta Z$. Clearly, when $\Delta Z$ is small relative to $Z_c$, the asymmetric division will result in larger overall population growth (see supplementary for further extensions of this model). Therefore, restricting cell division to one progenitor cell through the asymmetric segregation of the limiting resource can increase the overall population growth. Note also that as daughter cells are likely to require more resources to grow and divide compared with mother cells, mother-only proliferation should further maximize the number of cells given limited resources.

In addition to promoting overall population growth, the splitting of the population to mother and daughter cells may facilitate adaptation by implementing a risk-spreading strategy. Vacuoles are important for protecting cells against a variety of stresses, and indeed we find that mother cells better resist stress conditions, such as zinc toxicity or high osmolarity, known to require vacuolar function. At the same

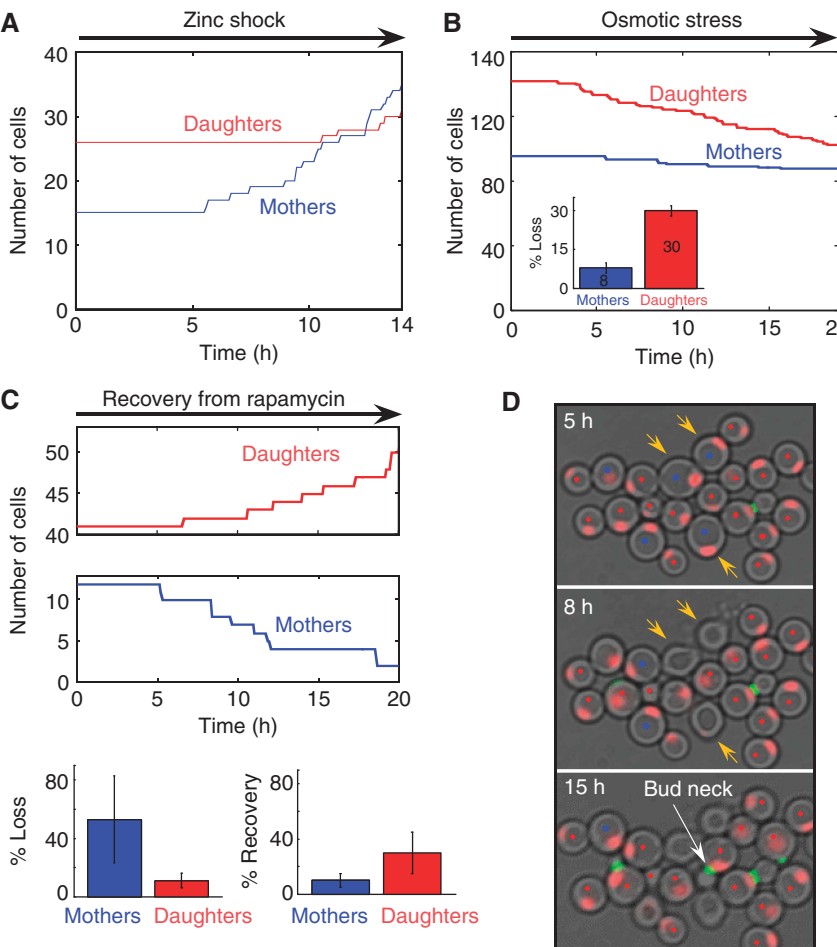

**Figure 7** Differential stress sensitivity of the dividing versus non-dividing cells. (**A**) Mothers are more resistant to zinc shock. Following a period of 7 h in low zinc conditions, the cells were exposed to extremely high-zinc conditions (SC medium supplemented with 9 mM $Zn^{2+}$). The shift to high zinc concentration leads to a period of growth arrest followed by a recovery of the mothers. The daughters start recovering only 4 h later. Similar results were obtained when cells deleted of the vacuole membrane zinc storage transporter, Zrc1, were shifted to rich conditions. (**B**) Mothers are more resistant to osmotic stress. Following a period of 8.5 h in low zinc, the cells were shifted ($t = 0$) to LZM with 150 mM NaCl and 1.2% Triton-X100. Shown is the decline in the number of mothers and daughters. Note that 30% of the daughters compared with only 8% of the mothers imploded and died (see Supplementary Movies 3a and b for visualization of dying cells) during the time of the stress. (**C**, **D**) Daughters are more resistant to rapamycin. Following a period of 12 h in low zinc, the cells were exposed to rapamycin (16 μM) for 16 h, and then were resupplied with rich medium. The number of mothers and their progeny versus daughters and their progeny in one field of view, during the recovery stage in rich medium, is shown in (C). Note that while most mothers died (exploded), the daughters managed to recover (see Supplementary Movie 3c). The bar plots (bottom) show the percentage of dying and recovering cells in both populations averaged over four fields of view ($\sim 100$ cells). Error bars represent the s.d. of different fields of view from the mean value. Snapshots of a typical colony showing exploding mothers next to budding daughters, are shown in (D): the three mother cells (blue dot) marked by yellow arrow at $t = 5$ h are shown to shrink/explode and to lose their nuclear marker (red) at $t = 8$ h. The budding daughters (red dot) can be identified by the appearance of the bud neck (green).

time, dividing cells are more sensitive to a variety of toxins or drugs that target proliferative mechanisms, and indeed we find that daughter cells better resist rapamycin, an example of such a drug. Therefore, at the population level, the splitting of phenotype may maximize protection in an uncertain environment.

Taken together, we propose that under conditions of metal limitation, budding yeast cells adopt a division cycle in which they prevent vacuole segregation by maintaining them within the mother cell. This restricts division to mother cells and may serve two purposes. First, it prevents the dilution of a limited resource thereby maximizing overall population growth. Second, it renders mothers and daughters resistant to different stress types, thereby implementing a risk-spreading strategy.

Notably, while previous studies have shown that mothers maintain damaged material to ensure rejuvenation of their progenies (Sinclair and Guarente, 1997; Aguilaniu *et al*, 2003), our data demonstrate that under stressful conditions mothers retain useful resources, which may be required for division and protection.

# Materials and methods

## Strains

The wild-type strain is based on Y8205 (Tong and Boone, 2006), and contains fusions of eGFP and mCherry to CDC10 and ACS2, respectively. Strains with fluorescent markers were made by transformation or by mating the wild-type strain with the corresponding a type

strain from the yeast deletion collection (EUROSCARF) (Giaever *et al*, 2002). See supplementary for a list of all strains used in this work. Deletion strains with florescent markers were made by transformation on the background of the corresponding strain from the deletion collection (EUROSCARF).

## Time-lapse microscopy

Growth of microcolonies was observed with fluorescence time-lapse microscopy at 30 °C using an Olympus IX-81-ZDC inverted microscope with a motorized stage (Prior). Image sets were acquired with a Hamamatsu ORCA-II-BT camera. Fluorescent proteins were detected using EXFO X-Cite light source and Chroma 49002ET-GFP and 49008ET-mCherry filter sets. Cells were observed in a microfluidic flow cell (Supplementary Figure 1a) connected to two microperfusion pumps, that enabled to grow the cells in a planar layer while simultaneously controlling their extracellular environment. ImagePro 6.3.1 (Media Cybernetics) was used to automate image acquisition, microscope control and pumps control.

## Growth conditions and cell preparation

Cells were pre-grown for 24 h in SC medium to $OD_{600}$ of about 0.5 and then were loaded to the flow cell. At $t = 0$, the medium was switched to a LZM and a set of bright field, red-fluorescence and green-fluorescence images was acquired for six fields of view, every 6 min. LZM were prepared as described in Zhao and Eide (1996), and had the following composition: 0.17% yeast nitrogen base without amino acids, $(NH_4)_2SO_4$, or zinc (ForMedium; CYN2301); 0.5% $(NH_4)_2SO_4$; 10 mM trisodium citrate, pH 4.2; 2% glucose; 1 mM $Na_2$ EDTA; and 1% adenine, histidine, leucine and tryptophan. The $MnCl_2$ and $FeCl_3$ concentrations in LZM were adjusted to final concentrations of 25 and 10 μM. Low copper media were prepared similar to LZM, with yeast nitrogen base without amino acids, $(NH_4)_2SO_4$ or copper (ForMedium; CYN0901) (instead of the YNB with no zinc), and addition of $ZnSO_4$ to final concentration of 1000 μM. Low iron media were prepared similar to LZM, with yeast nitrogen base without amino acids, $(NH_4)_2SO_4$ or iron (ForMedium; CYN1101) (instead of the YNB with no zinc), and addition of $ZnSO_4$ to final concentration of 1000 μM. Low-phosphate medium was prepared from YNB with no phosphate (ForMedium; CYN0804).

## Flow cell

In order to grow the cells in a planar layer while simultaneously controling their extracellular environment, we used a standard FCS2 Flow cell (Bioptechs) and improved it according to Charvin *et al* (2008), see Supplementary experimental procedures. As shown in Supplementary Figure 1a, the cells are confined between the PDMS layer and the membrane, while the medium is flowing above the membrane. In this way, the medium reaches the cell area without exerting too much force on the cells. The flow cell is then connected to two microperfusion pumps (Instech), which are controlled automatically by the computer via a D/A convener (measurement computing usb-3110), enabling to the changing of the medium during the experiment without perturbing the cells.

## Image analysis

Image analysis was performed by custom-written software in Matlab (Mathworks Inc.). We have analyzed movies from the end to the beginning, segmenting cells only in the last image and then tracking them to the first image. Nuclear markers facilitated the initial tracking and segmentation. Nuclear separation was identified by appearance of the nuclear marker in the daughter cell. Cell birth, defined by the bud neck disappearance, was identified as a significant decrease of the intensity of the bud neck marker in proximity (up to 30 min) to the nuclear separation. Cell size was estimated from the bright field images. Cells that were exploded were identified manually from the

bright field and the GFP images. In order to control for age differences in size distributions, we plotted the sizes as if all cells born in the same time.

## Supplementary information

## Acknowledgements

We thank Nathalie Balaban and Orit Gefen for advice in setting up the microscopic system, and the NCRR Yeast Resource Center, University of Washington, C Boone and M Schuldiner for plasmids and strains. We thank J Berman, B Shilo and members of our group for discussion and comments on the manuscript. This work was supported by the ERC, the Israel Science Foundation, Minerva, and the Hellen and Martin Kimmel award for innovative investigations.

*Author contributions:* The work presented here was carried out in collaboration between all authors. NA and NB defined and planned the research. NA set up the experimental setup, carried out the experiments and analyzed the data. IS wrote a major part of the image analysis program. MC constructed the strains used in this work. NA and NB interpreted the results and wrote the paper. All authors have contributed to, seen and approved the manuscript.

## Conflict of interest

The authors declare that they have no conflict of interest.

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
