## [Review Process File · Molecular Systems Biology]

Increasing population growth by asymmetric segregation of a limiting resource during cell division

Nurit Avraham, Ilya Soifer, Miri Carmi and Naama Barkai

Corresponding author: Naama Barkai, Department of Molecular Genetics, Weizmann Institute of Science

Review timeline:

Submission date:	26 September 2012
Editorial Decision:	23 October 2012
Revision received:	14 January 2013
Editorial Decision:	09 February 2013
Revision received:	23 February 2013
Accepted:	01 March 2013

Editors: Andrew Hufton / Thomas Lemberger

Transaction Report:

1st Editorial Decision

23 October 2012

Thank you again for submitting your work to Molecular Systems Biology. We have now heard back from the three referees who agreed to evaluate your manuscript. As you will see from the reports below, the referees find the topic of your study of potential interest. They raise, however, substantial concerns on your work, which, I am afraid to say, must preclude its publication in its present form.

The reviewers each recognized that the mother-only proliferation pattern observed in this work was intriguing. However, each also raised important concerns which will need to be conclusively addressed before this work would be suitable for publication. The editor would like to emphasize that some of these issues appear to require substantial additional experiment work to address. The concerns raised by Reviewer #2, in particular, seem to indicate that some of the central time microscopy experiments need to be repeated with a more controlled transition from high-zinc to low-zinc growth media, to assure that other media differences are not contributing to the observed proliferation pattern. Moreover, this reviewer also felt that zinc limitation as the causative agent behind this growth pattern should more directly demonstrated, and these concerns seem to fall somewhat in line with comments from Reviewer #3. Lastly, the reviewers felt that the modeling component of this work should be incorporated more directly into the Results section, and Reviewer #1 had some important concerns with the formulation of these models.

In addition, when preparing your revised manuscript, we ask that you consider supplying "source data" files for the Figures presented in this work, where applicable. In general, Molecular Systems Biology asks that authors supply the underlying numeric data for new experimental results. Figure source data is a new feature that allows these supplemental data to be more directly incorporated

into the main manuscript, and therefore more visible to interested readers (e.g. <http://tinyurl.com/365zpej>). Please see our Instructions of Authors for more details on preparation and formatting of figure source data (<http://www.nature.com/msb/authors/index.html#a3.4.3>).

Reviewer #1 (Remarks to the Author):

In the paper "Stress induced mother-only proliferation in budding yeast" by Avraham, et al., the authors report that budding yeast cells grown in low zinc adopt an asymmetric growth pattern that may optimize population growth in limiting conditions. This finding adds to the growing list of mechanisms through which microbes adapt to stressful environments. I believe that the phenotype is quite interesting, and the experiments are both beautiful and persuasive. However, the main text provides little intuition for the finding, since both the quantitative rationale and evolutionary implications of this growth strategy are buried in the supplement. Moreover, the models presented in the supplement, which are paramount for understanding the experimental results in a quantitative framework, are not convincingly presented or analyzed. I believe that the paper will be appropriate for publication in *Molecular Systems Biology* if the authors are able to respond to these concerns (detailed below).

Major comments

Broadly, the paper lacks any extended discussion of the implications of the asymmetric growth phenotype described experimentally. In particular, the "Discussion" section is extremely short, and the authors merely assert that (a) asymmetric growth maximizes population size and (b) the asymmetry could serve as the basis of a bet-hedging strategy (with data in Fig. S2). I believe that, at minimum, a description of some reasonable growth model should be included in the main text, since it would clarify the interpretation of the results.

Despite the importance of a model to provide insight to the phenomenon, I had some scientific concerns about the formulation of the models presented in the supplementary text. Thoughts regarding the specific models are detailed below.

Model I

For the case of asymmetric growth, the amount of resources per cell at a particular time is expressed as $Z_{(n+1)} = Z_n + \Delta Z$. In the case where $\Delta Z > 0$, this formulation seems to be problematic, as $Z \rightarrow \infty$ over time, predicting that the mother cell can divide without bound. Thus, the interpretation of the model presented seems incorrect.

Moreover, in the expression for N_{as} , I'm not sure that the absolute value arises naturally during the derivation. Of course, without the absolute value then the expression takes on a negative, non-physical value when $\Delta Z > 0$ and $Z_0 > Z_c$. However, the expression is not correct when $\Delta Z > 0$.

Indeed, the plot supplied by the authors seems to acknowledge these difficulties (and is not consistent with the authors' statement about the importance of whether ΔZ is larger or smaller than Z_c).

In addition to these concerns, I am also not sure if the symmetric division case was analyzed correctly. It seems that depending upon ΔZ , Z_0 , and Z_c the symmetric and asymmetric cases should give the same final number of cells (whereas in the plot the symmetric model always does worse). Indeed, a quick calculation leads me to think that the factor of 2 in the numerator of the expression should not be there.

In summary, a simple model such as Model I would be nice, but I am not convinced that the analysis is correct.

Model II

1. It was not obvious to me that the model needed to be stochastic. Indeed, the model may be easier to interpret if it is formulated deterministically.

2. In the authors' expression for $p(Z, \Delta_t)$ the parameter c "determines the width". Normally one would instead allow Δ_Z to determine the width and use c to specify the "amplitude" (or simply set $c = 0.5$ to indicate "saturation").
3. For the *whi5* mutant in low zinc, growth seems to be linear and not exponential (orange line on Fig. S7(c) and S7(d)), even though divisions are symmetric.
4. More generally, this model is sufficiently complicated that it is difficult to get intuition for the mechanism of why the asymmetric division might be superior.

I thought that the authors would assume (reasonably) that it would take more resources for a daughter cell to grow and divide as compared to the mother cell. In this case, mother-only proliferation should maximize the number of cells given limited resources. I do not feel that the authors necessarily need to include this mechanism in their model, but it seems likely to be relevant.

Minor comments

1. The data in Supplementary Fig. 2 is quite nice, especially given that mothers are favored in some conditions, while daughters are favored in others. It might be nice to elaborate on the discussion of this feature in the main text. I would even recommend including this figure in the main text.
2. In Figure 3A, it is not obvious how the figure takes the age difference between mothers and daughters into account in the size distribution. Was there some normalization?
3. Labels for the subplots within each figure (a, b, c, etc.) should be in a consistent spot (i.e. top left of the subplot).
4. There are a couple of spelling mistakes in Fig. 2 ("phosphat" instead of "phosphate").
5. The introduction is quite short and doesn't quite motivate the subject. Why study zinc (as opposed to glucose, for instance, in which growth is also purportedly asymmetric)?

Reviewer #2 (Remarks to the Author):

This manuscript raises the very intriguing hypothesis that yeast cells under nutrient limitation transition from exponential growth to a linear growth phase by restricting cell divisions to mother cells. This pattern of growth correlates with the altered inheritance of the vacuole that occurs under these conditions. Given that the vacuole is the primary site of zinc storage, it appears that conservation of the zinc store in a single cell is the goal. This strategy would allow for production of daughter cells while maintaining the cell division capacity of a subset of the population such that growth can be restored quickly after nutrient resupply. The work is very interesting but some deficiencies were noted that compromised the authors' conclusions.

- 1) First, it is unclear what "rich" medium is; YPD, as is usually meant by the term, or SC?.
- 2) Second, when the authors transfer cells from SC to low zinc (e.g. Figure 1b), there are many changes in growth conditions in addition to zinc availability. SC (i.e. "synthetic complete") normally contains a large number of supplemented amino acids etc while the low zinc medium used only contains adenine, histidine, leucine, and tryptophan (and uracil? - not mentioned) as supplements. Also, SC medium is not pH-buffered while the low zinc medium is pH-buffered by citrate at a relatively low pH. Do these other factors contribute to the change in division pattern that occurs or is it really an effect of changing zinc status? Experiments comparing zinc-replete and low zinc conditions should be added in which these other variables are controlled. For example, by comparing low zinc medium with 10 μ M Zn with low zinc medium made replete by adding 100 μ M or more zinc.
- 3) Similarly, the experiments with low copper and low iron media as described in Figure 2 should be repeated comparing the low copper/iron conditions with conditions in which those metals are added back to that same medium.
- 4) The concentrations of Mn and Fe in the medium used are extremely high (25 and 10 mM respectively as stated in both the materials and methods and the supplemental information) which raises the great concern that the effect is due to metal toxicity rather than zinc (or copper or iron) deficiency. These may be just typographical errors but, if not, control experiments should be performed at lower metal concentrations (25 and 10 μ M were the concentrations used by Zhao and Eide) to ensure that this is not due to Mn or Fe toxicity.

- 5) The key component of the model is that the mother-only division strategy is to maintain the zinc storage pool in the mother cells. This hypothesis should be further tested. Specifically, what effect does the size of the vacuolar zinc pool have on the timing of the transition? Does an increased zinc pool delay the transition while a decreased pool cause it to occur under less severe zinc deficiency? Increased zinc pools can be obtained by pregrowing the cells in SC with 1 mM Zn. Cells with decreased zinc pools can be generated using a *zrc1 cot1* mutant.
- 6) Are the low phosphate conditions in Figure 2 truly low phosphate? That is, are the cells growing more slowly than phosphate-replete cells?
- 7) Figure 4A, C, E. The y-axis is labeled as "Vacuole/cell size" which isn't clear to me; does a value of 1.0 mean the entire cell is vacuole? Given the error bars, it appears that some values were greater than 1.0 so this interpretation can't be correct.
- 8) Figure 5C and D are currently mislabeled "number of cells".
- 9) I would add mention of the low copper and low iron effects to the abstract as this is not a zinc-specific effect.
- 10) Figure 2e and f- "phosphate"
- 11) Figure 2e and f legend. "...for low phosphate conditions..."
- 12) Figure 3a legend. It would help to clarify that the comparison is of mother and daughter cells that are 7 hours old.
- 13) Supplemental Figure 1 legend: the descriptions of panels b and c are switched.
- 14) Supplemental Figure 5 legend: FluoZin-1 detects labile zinc only and not "cellular zinc" (the latter suggesting total zinc). Also, the supplemental experimental procedures mentions FuraZin-1 and not FluoZin-1.

Reviewer #3 (Remarks to the Author):

This paper reports the observation that when *Saccharomyces cerevisiae* (budding yeast) cells are exposed to environments containing low concentrations of essential metals (zinc, copper and iron) mother cells continue to produce daughter cells; however, their daughter cells are unable to divide. When the G1/S transition inhibitor *WHI5* is deleted this asymmetry is lost and both mothers and daughters divide in low zinc environments. However, under these conditions *whi5* mother and daughter cells progressively slow in growth resulting in reduced overall population growth rate. The authors show that the growth behavior is correlated with vacuole size and inheritance.

The regulation of cell growth, including the underlying mechanisms, sources of heterogeneity and evolutionary implications is an important area of research. This paper represents a valuable contribution to the field and capitalizes on the unique capabilities of microfluidic devices and real time imaging of cell growth. The experiments are well executed and the data support the conclusions of the authors. I recommend publication after the authors have addressed the following comments.

1. The authors refer to "limiting zinc conditions". The term "limiting" is not defined and I would expect to see confirmation that final culture density is a function of zinc concentration to justify the claim that the concentration is indeed limiting. I suggest they use "low".
2. In Figure 1a and 1b it is unclear what the y-axis represents. The authors are measuring microcolonies of cells as shown in Figure 1c. How do they get values in the 100s (Fig 1a) or 10s (Fig 1b) and why are they different? Similarly, in Figure 2 it is unclear if these plots are aggregates of multiple microcolonies or a single microcolony.
3. I disagree that this "...proliferation pattern represents a general response to nutrient limitation..." As the authors show this behavior is unique to low concentrations of essential metals and does not

occur in phosphate limitation.

4. The authors claim that the daughter cells are "largely devoid of their vacuoles". It seems unlikely (and not proven) that they would have no vacuoles, and more likely that the vacuoles are simply smaller and therefore hard to detect.

5. I find it odd that the authors only introduce the mathematical model in the discussion. Why not include it in the main text?

6. The title is misleading as the phenomenon is only observed in a very specific type of stress (and stress is a very subjective classification). Maybe "Budding yeast adopts a mother only proliferation pattern in low metal environments".

1st Revision - authors' response

14 January 2013

Reviewer #1 (Remarks to the Author):

In the paper "Stress induced mother-only proliferation in budding yeast" by Avraham, et al., the authors report that budding yeast cells grown in low zinc adopt an asymmetric growth pattern that may optimize population growth in limiting conditions. This finding adds to the growing list of mechanisms through which microbes adapt to stressful environments. I believe that the phenotype is quite interesting, and the experiments are both beautiful and persuasive. However, the main text provides little intuition for the finding, since both the quantitative rationale and evolutionary implications of this growth strategy are buried in the supplement. Moreover, the models presented in the supplement, which are paramount for understanding the experimental results in a quantitative framework, are not convincingly presented or analyzed. I believe that the paper will be appropriate for publication in Molecular Systems Biology if the authors are able to respond to these concerns (detailed below).

We thank the reviewer for those comments. As suggested, we extended the introduction and discussion and included Model #1 in main text (corrected for the properly noted mistakes/typos).

Major comments

Broadly, the paper lacks any extended discussion of the implications of the asymmetric growth phenotype described experimentally. In particular, the "Discussion" section is extremely short, and the authors merely assert that (a) asymmetric growth maximizes population size and (b) the asymmetry could serve as the basis of a bet-hedging strategy (with data in Fig. S2). I believe that, at minimum, a description of some reasonable growth model should be included in the main text, since it would clarify the interpretation of the results.

The discussion section was extended and model I included in main text.

Despite the importance of a model to provide insight to the phenomenon, I had some scientific concerns about the formulation of the models presented in the supplementary text. Thoughts regarding the specific models are detailed below.

Model I

For the case of asymmetric growth, the amount of resources per cell at a particular time is expressed as $Z(n+1) = Z_n + \Delta Z$. In the case where $\Delta Z > 0$, this formulation seems to be problematic, as $Z \rightarrow \infty$ over time, predicting that the mother cell can divide without bound. Thus, the interpretation of the model presented seems incorrect.

The model is indeed design to address only the case of limitation, where $\Delta Z < 0$. In non-limiting conditions, the cell has means to prevent uncontrolled accumulation. For example, any first-order degradation will suffice to maintain the limiting factor at fixed concentration.

We now emphasize this assumption in the main text when introducing the model.

Moreover, in the expression for N_{as} , I'm not sure that the absolute value arises naturally during the derivation. Of course, without the absolute value then the expression takes on a negative, non-physical value when $\Delta Z > 0$ and $Z_0 > Z_c$. However, the expression is not correct when $\Delta Z > 0$. Indeed, the plot supplied by the authors seems to acknowledge these difficulties (and is not consistent with the authors' statement about the importance of whether ΔZ is larger or smaller than Z_c).

We thank the reviewer for this comment. The absolute value was removed and the conditions where this expression holds are now specified.

In addition to these concerns, I am also not sure if the symmetric division case was analyzed correctly. It seems that depending upon ΔZ , Z_0 , and Z_c the symmetric and asymmetric cases should give the same final number of cells (whereas in the plot the symmetric model always does worse). Indeed, a quick calculation leads me to think that the factor of 2 in the numerator of the expression should not be there.

This is correct. We apologize for this mistake.

In summary, a simple model such as Model I would be nice, but I am not convinced that the analysis is correct.

We believe that this is corrected now.

Model II

1. It was not obvious to me that the model needed to be stochastic. Indeed, the model may be easier to interpret if it is formulated deterministically.

2. In the authors' expression for $p(Z, \Delta Z)$ the parameter c "determines the width". Normally one would instead allow ΔZ to determine the width and use c to specify the "amplitude" (or simply set $c = 0.5$ to indicate "saturation").

We agree with both comments.

The purpose of this "Model II" is to extend "Model I" to a more realistic setting. For this, we included three additional features that were not considered in Model I: First, we allowed accumulation of the limiting factor during the cell cycle (in Model I, we only assumed depletion by dilution or utilization). Second, we allowed different levels of asymmetric segregation (in Model I we considered only the limiting cases whereby division is either fully symmetric or fully asymmetric). Third, we allowed stochasticity in the cell-division threshold (in Model I, division was allowed only if the level of limited resource exceeds a fixed threshold). The purpose was to demonstrate feasibility of accounting for the actual dynamics observed, while the intuition was given already in the first model.

This motivation for Model II was not properly explained before and we now better emphasize this point.

Specifically, for the point of stochasticity, we now simulate also this extended model also in the deterministic case (see Supplementary Figure 6e and 6f). The results are the same.

3. For the *whi5* mutant in low zinc, growth seems to be linear and not exponential (orange line on Fig. S7(c) and S7(d)), even though divisions are symmetric.

As we explain in the text, the deviation from exponential growth is due to the progressive increase in cell cycle times of the mutants following the transfer to low-zinc conditions. To avoid confusion, those figures were replaced by figures that provide a better intuition.

4. More generally, this model is sufficiently complicated that it is difficult to get intuition for the mechanism of why the asymmetric division might be superior.

We tried to provide intuition by presenting the simple Model I, while the purpose of Model II was to include additional properties perhaps making it more realistic. We failed to explain this in the previous version, but now emphasize this point and hope this is clearer.

I thought that the authors would assume (reasonably) that it would take more resources for a daughter cell to grow and divide as compared to the mother cell. In this case, mother-only proliferation should maximize the number of cells given limited resources. I do not feel that the authors necessarily need to include this mechanism in their model, but it seems likely to be relevant.

We agree. We added this point to the discussion.

Minor comments

1. The data in Supplementary Fig. 2 is quite nice, especially given that mothers are favored in some conditions, while daughters are favored in others. It might be nice to elaborate on the discussion of this feature in the main text. I would even recommend including this figure in the main text.

This figure is now included in the main text as Figure 7, and we elaborated our discussion of this bet-hedging phenotype.

2. In Figure 3A, it is not obvious how the figure takes the age difference between mothers and daughters into account in the size distribution. Was there some normalization?

The age difference was taken into account in the inset of figure 3a. Specifically, in the inset we considered only cells that are precisely seven hours 'old' and compare the sizes of mothers and daughters. (Seven was chosen just for convenience, other ages give the same results)

3. Labels for the subplots within each figure (a, b, c, etc.) should be in a consistent spot (i.e. top left of the subplot).

Corrected- all labels are now at the top left corner of the subplots.

4. There are a couple of spelling mistakes in Fig. 2 ("phosphat" instead of "phosphate").

This spelling mistake was corrected (Figure 1e in the revised manuscript).

5. The introduction is quite short and doesn't quite motivate the subject. Why study zinc (as opposed to glucose, for instance, in which growth is also purportedly asymmetric)?

We extended the introduction to better motivate our choice of system (s). We would like to note, though that we began studying zinc with another question in mind and noted this interesting phenotype, rather than looking for it to occur specifically in zinc.

Reviewer #2 (Remarks to the Author):

This manuscript raises the very intriguing hypothesis that yeast cells under nutrient limitation transition from exponential growth to a linear growth phase by restricting cell divisions to mother cells. This pattern of growth correlates with the altered inheritance of the vacuole that occurs under these conditions. Given that the vacuole is the primary site of zinc storage, it appears that conservation of the zinc store in a single cell is the goal. This strategy would allow for production of daughter cells while maintaining the cell division capacity of a subset of the population such that

growth can be restored quickly after nutrient resupply. The work is very interesting but some deficiencies were noted that compromised the authors' conclusions.

1) First, it is unclear what "rich" medium is; YPD, as is usually meant by the term, or SC?

In our case, "rich medium" means SC. This is now explained the first time "rich medium" is mentioned in the text

2) Second, when the authors transfer cells from SC to low zinc (e.g. Figure 1b), there are many changes in growth conditions in addition to zinc availability. SC (i.e. "synthetic complete") normally contains a large number of supplemented amino acids etc while the low zinc medium used only contains adenine, histidine, leucine, and tryptophan (and uracil? - not mentioned) as supplements. Also, SC medium is not pH-buffered while the low zinc medium is pH-buffered by citrate at a relatively low pH. Do these other factors contribute to the change in division pattern that occurs or is it really an effect of changing zinc status? Experiments comparing zinc-replete and low zinc conditions should be added in which these other variables are controlled. For example, by comparing low zinc medium with 10 μ M Zn with low zinc medium made replete by adding 100 μ M or more zinc.

We performed the requested experiment (summarized now in Supplementary Figure 2a and 2b) and observed the same phenotype as that when transferring cells from SC to low-zinc media.

3) Similarly, the experiments with low copper and low iron media as described in Figure 2 should be repeated comparing the low copper/iron conditions with conditions in which those metals are added back to that same medium.

We performed the requested experiment (summarized now in Supplementary Figure 2) and observed the same phenotype as that when transferring cells from SC to the respective limiting media.

4) The concentrations of Mn and Fe in the medium used are extremely high (25 and 10 mM respectively as stated in both the materials and methods and the supplemental information) which raises the great concern that the effect is due to metal toxicity rather than zinc (or copper or iron) deficiency. These may be just typographical errors but, if not, control experiments should be performed at lower metal concentrations (25 and 10 μ M were the concentrations used by Zhao and Eide) to ensure that this is not due to Mn or Fe toxicity.

We apologize for this typo. The concentrations used were 25 μ M and 10 μ M.

*5) The key component of the model is that the mother-only division strategy is to maintain the zinc storage pool in the mother cells. This hypothesis should be further tested. Specifically, what effect does the size of the vacuolar zinc pool have on the timing of the transition? Does an increased zinc pool delay the transition while a decreased pool cause it to occur under less severe zinc deficiency? Increased zinc pools can be obtained by pregrowing the cells in SC with 1 mM Zn. Cells with decreased zinc pools can be generated using a *zrc1 cot1* mutant.*

We performed the two suggested experiments. First, we examined the *zrc1 cot1* double-deletion mutants transferred to severe (10 μ M) or less severe (50 μ M) zinc limitation. As predicted, the double mutants were more sensitive: the segregated phenotype began earlier, and mothers underwent a smaller number of divisions. These results are now discussed in main text, and presented in the Figure 6.

We also performed the second suggested experiment of pre-growing the cells in SC with 1mM Zn. However, we couldn't observe an effect of this per-grown phase. This result is difficult to interpret as we do not know the amount of additional vacuole zinc under those conditions, and the possible changes in utilization mechanism.

6) *Are the low phosphate conditions in Figure 2 truly low phosphate? That is, are the cells growing more slowly than phosphate-replete cells?*

We think so. In the first few generations after the transition, the cells grow about 1.6 times slower than cells growing when Pi is present in the medium.

7) *Figure 4A, C. E. The y-axis is labeled as "Vacuole/cell size" which isn't clear to me; does a value of 1.0 mean the entire cell is vacuole? Given the error bars, it appears that some values were greater than 1.0 so this interpretation cant be correct.*

This interpretation is correct, value of 1 means that the entire cell is vacuole. We now changed the scale and it can be shown that there are only 3 cells that there value was close to one. In these cases the upper side of the error bar is of course limited by this value. We changed the limits accordingly (figure 5d in the revised version)

8) *Figure 5C and D are currently mislabeled "number of cells".*

Corrected

9) *I would add mention of the low copper and low iron effects to the abstract as this is not a zinc-specific effect.*

Added

10) *Figure 2e and f- "phosphate"*

Corrected

11) *Figure 2e and f legend. "...for low phosphate conditions..."*

Corrected

12) *Figure 3a legend. It would help to clarify that the comparison is of mother and daughter cells that are 7 hours old.*

This is now clarified in the figure caption

13) *Supplemental Figure 1 legend: the descriptions of panels b and c are switched.*

Corrected

14) *Supplemental Figure 5 legend: FluoZin-1 detects labile zinc only and not "cellular zinc" (the latter suggesting total zinc). Also, the supplemental experimental procedures mentions FuraZin-1 and not FluoZin-1.*

Corrected

Reviewer #3 (Remarks to the Author):

*This paper reports the observation that when *Saccharomyces cerevisiae* (budding yeast) cells are exposed to environments containing low concentrations of essential metals (zinc, copper and iron)*

mother cells continue to produce daughter cells; however, their daughter cells are unable to divide. When the G1/S transition inhibitor WHI5 is deleted this asymmetry is lost and both mothers and daughters divide in low zinc environments. However, under these conditions whi5 mother and daughter cells progressively slow in growth resulting in reduced overall population growth rate. The authors show that the growth behavior is correlated with vacuole size and inheritance.

The regulation of cell growth, including the underlying mechanisms, sources of heterogeneity and evolutionary implications is an important area of research. This paper represents a valuable contribution to the field and capitalizes on the unique capabilities of microfluidic devices and real time imaging of cell growth. The experiments are well executed and the data support the conclusions of the authors. I recommend publication after the authors have addressed the following comments.

1. The authors refer to "limiting zinc conditions". The term "limiting" is not defined and I would expect to see confirmation that final culture density is a function of zinc concentration to justify the claim that the concentration is indeed limiting. I suggest they use "low".

'Limiting zinc conditions' was changed to 'low zinc'.

2. In Figure 1a and 1b it is unclear what the y-axis represents. The authors are measuring microcolonies of cells as shown in Figure 1c. How do they get values in the 100s (Fig 1a) or 10s (Fig 1b) and why are they different? Similarly, in Figure 2 it is unclear if these plots are aggregates of multiple microcolonies or a single microcolony.

This is now better explained in the figure caption. The y axis represents the number of cells within a field of view. In the case of rich conditions the growth is exponential and therefore the number of cells is of the order hundreds. In low zinc they grow linearly reaching lower cell numbers (order tens).

3. I disagree that this "...proliferation pattern represents a general response to nutrient limitation..." As the authors show this behavior is unique to low concentrations of essential metals and does not occur in phosphate limitation.

The text was modified to emphasize that we see the mother-restricted division pattern in only some stresses and not in others.

4. The authors claim that the daughter cells are "largely devoid of their vacuoles". It seems unlikely (and not proven) that they would have no vacuoles, and more likely that the vacuoles are simply smaller and therefore hard to detect.

"Were largely devoid of their vacuole" was changed to "received very small vacuoles".

5. I find it odd that the authors only introduce the mathematical model in the discussion. Why not include it in the main text?

Model I is now included in main text.

6. The title is misleading as the phenomenon is only observed in a very specific type of stress (and stress is a very subjective classification). Maybe "Budding yeast adopts a mother only proliferation pattern in low metal environments".

Title was changed.

Additional changes

The figures were changed in the following way:

Figure 1 and 2 were reorganized.

Figure 6 was added

Supplementary Figure 2 was moved to the revised manuscript as Figure 7

2nd Editorial Decision

09 February 2013

Thank you again for submitting your work to Molecular Systems Biology. We have now heard back from the two referees who accepted to evaluate the revised study. As you will see, they are now supportive and we will be able to accept your manuscript for publication pending the following minor points.

Source data for figures:

Molecular Systems Biology strongly encourages authors to upload the 'source data' that were used to generate figures—for example, tables of individual numerical values and measurements. These files are separate from the traditional supplementary information files and are submitted using the "figure source data" option in the tracking system. Source data are directly linked to specific figure panels so that interested readers can directly download the associated 'source data' (see, for example, <http://tinyurl.com/365zpej>), for the purpose of alternative visualization, re-analysis or integration with other data. Additional formatting guidelines for 'source data' are available for download [<http://www.nature.com/msb/authors/source-data.pdf>].

Reviewer #1 (Remarks to the Author):

The authors have addressed almost all of my concerns. I believe that this study represents a significant advance in our understanding of microbial responses to stressful environments, and should be published in MSB. Moreover, the data are beautiful (in particular the visualization of exponential vs linear population growth is fantastic).

One comment on the model that I believe is important. The authors write:

"This simple model can easily be extended to perhaps a more realistic situation where a fraction Z of the limiting resource is lost at each division"

The authors are not referring to a fraction ΔZ that is lost, but rather an amount ΔZ that is lost. I believe that the sentence should read:

"This simple model can easily be extended to perhaps a more realistic situation where an amount Z of the limiting resource is lost at each division"

Below I have a few suggestions that might make the paper even stronger.

At the top of pg 6: "A fluorescent GFP reporter driven by the ZRT1 promoter was strongly induced in both mother and daughter cells, and this induction was indistinguishable between the dividing and non-dividing cells (See Supplementary Figure 1d)."

Given that the mother cells are growing / dividing and the daughter cells are not, this means that all proteins, including GFP, will be diluted in the mother cell. Doesn't this mean that protein expression is higher in the mother cell than in the daughter cell?

The paragraph at the beginning of pg. 9 of the supplement ("Clearly, under a broad set of parameters...") is a little brief. I also remember there being some plots for Model I, which were nice for illustrating the results.

Theoretical studies suggest that when conditions fluctuate in unpredictable way, ...

Theoretical studies suggest that when conditions fluctuate in an unpredictable way, ...

In the budding yeast, the cell division is inherently asymmetric, and this asymmetry is amplified under nutrient limitations

Cell division is inherently asymmetric in budding yeast, and this asymmetry is amplified under nutrient limitations

Whether this population splitting into mothers and daughters provides some advantage for population growth or survival, is poorly understood.

However, it is not known whether this population splitting into mothers and daughters provides some advantage for population growth or survival.

To this end, we used an automated video microscopy to...

To this end, we used automated video microscopy to ...

of one of the transition metals, zinc, iron or copper

of one of the transition metals zinc, iron or copper

contrasting its symmetric segregation between mother and daughter when nutrient is abundant.

in contrast to its symmetric segregation between mother and daughter when nutrients are abundant.

the increase in cells number became linear in time

the increase in cell number became linear in time

continued to divide in low zinc for up to eight times

continued to divide in low zinc up to eight times

"While mother cells better resist stresses requiring vacuole protection (Li et al., 2012), daughter cells may better resist stresses, or drugs that affect dividing cells." It seems that the latter part of this sentence was not finished. Perhaps:

"While mother cells better resist stresses requiring vacuole protection (Li et al., 2012), daughter cells may better resist other stresses, in particular drugs that affect dividing cells."

We observe that when dividing in low zinc, mother cells maintain most of the vacuoles, contrasting the proportioned vacuole distribution between mother and daughters seen in rich media

We observe that when dividing in low zinc, mother cells maintain most of the vacuoles, in contrast to the more equal vacuole distribution between mother and daughters seen in rich media

replenished, such strategy maximizes population growth

replenished, such a strategy maximizes population growth

Taken together, we propose that under conditions of metal-limitations budding yeast cells adapt a division cycle in which they prevent vacuole segregation but maintain them within the mother cell.

(multiple changes)

Taken together, we propose that under conditions of metal-limitation budding yeast cells adopt a division cycle in which they prevent vacuole segregation by maintaining them within the mother cell.

This restricts division to mother cells and may serve for two purposes.

This restricts division to mother cells and may serve two purposes.

previous studies have shown that mothers maintain the damaged material to ensure

previous studies have shown that mothers maintain damaged material to ensure

demonstrates that under stressful conditions, mothers retain useful resources

demonstrates that under stressful conditions mothers retain useful resources

2nd Revision - authors' response

23 February 2013

Reviewer #1 (Remarks to the Author):

In the paper "Stress induced mother-only proliferation in budding yeast" by Avraham, et al., the authors report that budding yeast cells grown in low zinc adopt an asymmetric growth pattern that may optimize population growth in limiting conditions. This finding adds to the growing list of mechanisms through which microbes adapt to stressful environments. I believe that the phenotype is quite interesting, and the experiments are both beautiful and persuasive. However, the main text provides little intuition for the finding, since both the quantitative rationale and evolutionary implications of this growth strategy are buried in the supplement. Moreover, the models presented in the supplement, which are paramount for understanding the experimental results in a quantitative framework, are not convincingly presented or analyzed. I believe that the paper will be appropriate for publication in Molecular Systems Biology if the authors are able to respond to these concerns (detailed below).

We thank the reviewer for those comments. As suggested, we extended the introduction and discussion and included Model #1 in main text (corrected for the properly noted mistakes/typos).

Major comments

Broadly, the paper lacks any extended discussion of the implications of the asymmetric growth phenotype described experimentally. In particular, the "Discussion" section is extremely short, and the authors merely assert that (a) asymmetric growth maximizes population size and (b) the asymmetry could serve as the basis of a bet-hedging strategy (with data in Fig. S2). I believe that, at minimum, a description of some reasonable growth model should be included in the main text, since it would clarify the interpretation of the results.

The discussion section was extended and model I included in main text.

Despite the importance of a model to provide insight to the phenomenon, I had some scientific concerns about the formulation of the models presented in the supplementary text. Thoughts regarding the specific models are detailed below.

Model I

For the case of asymmetric growth, the amount of resources per cell at a particular time is expressed as $Z_{(n+1)} = Z_n + \Delta Z$. In the case where $\Delta Z > 0$, this formulation seems to be problematic, as $Z \rightarrow \infty$ over time, predicting that the mother cell can divide without bound.

Thus, the interpretation of the model presented seems incorrect.

The model is indeed design to address only the case of limitation, where $\Delta Z < 0$. In non-limiting conditions, the cell has means to prevent uncontrolled accumulation. For example, any first-order degradation will suffice to maintain the limiting factor at fixed concentration.

We now emphasize this assumption in the main text when introducing the model.

Moreover, in the expression for N as, I'm not sure that the absolute value arises naturally during the derivation. Of course, without the absolute value then the expression takes on a negative, non-physical value when $\Delta Z > 0$ and $Z_0 > Z_c$. However, the expression is not correct when $\Delta Z > 0$. Indeed, the plot supplied by the authors seems to acknowledge these difficulties (and is not consistent with the authors' statement about the importance of whether ΔZ is larger or smaller than Z_c).

We thank the reviewer for this comment. The absolute value was removed and the conditions where this expression holds are now specified.

In addition to these concerns, I am also not sure if the symmetric division case was analyzed correctly. It seems that depending upon ΔZ , Z_0 , and Z_c the symmetric and asymmetric cases should give the same final number of cells (whereas in the plot the symmetric model always does worse). Indeed, a quick calculation leads me to think that the factor of 2 in the numerator of the expression should not be there.

This is correct. We apologize for this mistake.

In summary, a simple model such as Model I would be nice, but I am not convinced that the analysis is correct.

We believe that this is corrected now.

1. It was not obvious to me that the model needed to be stochastic. Indeed, the model may be easier to interpret if it is formulated deterministically.

2. In the authors' expression for $p(Z, \Delta t)$ the parameter c "determines the width". Normally one would instead allow ΔZ to determine the width and use c to specify the "amplitude" (or simply set $c = 0.5$ to indicate "saturation").

We agree with both comments.

The purpose of this "Model II" is to extend "Model" I to more realistic setting. For this, we included three additional features that were not considered in Model I: First, we allowed accumulation of the limiting factor during the cell cycle (in Model I, we only assumed depletion by dilution or utilization). Second, we allowed different levels of asymmetric segregation (in Model I we considered only the limiting cases whereby division is either fully symmetric or fully asymmetric). Third, we allowed stochasticity in the cell-division threshold (in Model I, division was allowed only if the level of limited resource exceeds a fixed threshold). The purpose was to demonstrate feasibility of accounting for the actual dynamics observed, while the intuition was given already in the first model.

This motivation for Model II was not properly explained before and we now better emphasize this point.

Specifically, for the point of stochasticity, we now simulate also this extended model also in the deterministic case (see SuppFigures #). The results are the same.

3. For the *whi5* mutant in low zinc, growth seems to be linear and not exponential (orange line on Fig. S7(c) and S7(d)), even though divisions are symmetric.

As we explain in the text, the deviation from exponential is due to the progressive increase in cell cycle times of the mutants following the transfer to low-zinc conditions. To avoid confusion, those figures were replaced by figures that provide a better intuition.

4. More generally, this model is sufficiently complicated that it is difficult to get intuition for the mechanism of why the asymmetric division might be superior.

We tried to provide intuition by presenting the simple Model I, while the purpose of Model II was to include additional properties perhaps making it more realistic. We failed to explain this in the previous version, but now emphasize this point and hope this is clearer.

I thought that the authors would assume (reasonably) that it would take more resources for a daughter cell to grow and divide as compared to the mother cell. In this case, mother-only proliferation should maximize the number of cells given limited resources. I do not feel that the authors necessarily need to include this mechanism in their model, but it seems likely to be relevant.

We agree. We added this point to the discussion.

Minor comments

1. The data in Supplementary Fig. 2 is quite nice, especially given that mothers are favored in some conditions, while daughters are favored in others. It might be nice to elaborate on the discussion of this feature in the main text. I would even recommend including this figure in the main text.

This figure is now included in main text as Figure 7, and we elaborated our discussion of this bet-hedging phenotype.

2. In Figure 3A, it is not obvious how the figure takes the age difference between mothers and daughters into account in the size distribution. Was there some normalization?

The age difference was taken into account in the inset of figure 3a. Specifically, in the inset we considered only cells that are precisely seven hours 'old' and compare the sizes of mothers and daughters. (Seven was chosen just for convenience, other ages give the same results)

3. Labels for the subplots within each figure (a, b, c, etc.) should be in a consistent spot (i.e. top left of the subplot).

Corrected- all labels are now at the top left corner of the subplots.

4. There are a couple of spelling mistakes in Fig. 2 ("phosphat" instead of "phosphate").

This spelling mistake was corrected (Figure 1e in the revised manuscript).

5. The introduction is quite short and doesn't quite motivate the subject. Why study zinc (as opposed to glucose, for instance, in which growth is also purportedly asymmetric)?

We extended the introduction to better motivate our choice of system (s). We would like to note, though that we began studying zinc with another question in mind and noted this interesting phenotype, rather than looking for it to occur specifically in zinc.

Reviewer #2 (Remarks to the Author):

This manuscript raises the very intriguing hypothesis that yeast cells under nutrient limitation transition from exponential growth to a linear growth phase by restricting cell divisions to mother cells. This pattern of growth correlates with the altered inheritance of the vacuole that occurs under these conditions. Given that the vacuole is the primary site of zinc storage, it appears that conservation of the zinc store in a single cell is the goal. This strategy would allow for production of daughter cells while maintaining the cell division capacity of a subset of the population such that growth can be restored quickly after nutrient resupply. The work is very interesting but some deficiencies were noted that compromised the authors' conclusions.

1) First, it is unclear what "rich" medium is; YPD, as is usually meant by the term, or SC?.

In our case, “rich medium” means SC. This is explained the first time “rich medium” is mentioned in the text

2) Second, when the authors transfer cells from SC to low zinc (e.g. Figure 1b), there are many changes in growth conditions in addition to zinc availability. SC (i.e. "synthetic complete") normally contains a large number of supplemented amino acids etc while the low zinc medium used only contains adenine, histidine, leucine, and tryptophan (and uracil? - not mentioned) as supplements. Also, SC medium is not pH-buffered while the low zinc medium is pH-buffered by citrate at a relatively low pH. Do these other factors contribute to the change in division pattern that occurs or is it really an effect of changing zinc status? Experiments comparing zinc-replete and low zinc conditions should be added in which these other variables are controlled. For example, by comparing low zinc medium with 10 uM Zn with low zinc medium made replete by adding 100 uM or more zinc.

We performed the requested experiment (summarized now in supplementary fig. 2a and 2b) and observed the same phenotype as that when transferring cells from SC to low-zinc media.

3) Similarly, the experiments with low copper and low iron media as described in Figure 2 should be repeated comparing the low copper/iron conditions with conditions in which those metals are added back to that same medium.

We performed the requested experiment (summarized now in supplementary fig. 2) and observed the same phenotype as that when transferring cells from SC to the respective limiting media.

4) The concentrations of Mn and Fe in the medium used are extremely high (25 and 10 mM respectively as stated in both the materials and methods and the supplemental information) which raises the great concern that the effect is due to metal toxicity rather than zinc (or copper or iron) deficiency. These may be just typographical errors but, if not, control experiments should be performed at lower metal concentrations (25 and 10 uM were the concentrations used by Zhao and Eide) to ensure that this is not due to Mn or Fe toxicity.

We apologize for this typo. The concentrations used were 25 uM and 10 uM.

5) *The key component of the model is that the mother-only division strategy is to maintain the zinc storage pool in the mother cells. This hypothesis should be further tested. Specifically, what effect does the size of the vacuolar zinc pool have on the timing of the transition? Does an increased zinc pool delay the transition while a decreased pool cause it to occur under less severe zinc deficiency? Increased zinc pools can be obtained by pregrowing the cells in SC with 1 mM Zn. Cells with decreased zinc pools can be generated using a *zrc1 cot1* mutant.*

We performed the two suggested experiments. First, we examined the *zrc1 cot1* double-deletion mutants transferred to severe (10 μ M) or less severe (50 μ M) zinc limitation. As predicted, the double mutants were more sensitive: the segregated phenotype began earlier, and mothers underwent a smaller number of divisions. These results are now discussed in main text, and presented in the Figure 6.

We also performed the second suggested experiment of pre-growing the cells in SC with 1mM Zn. However, we couldn't observe an effect of this per-grown phase. This result is difficult to interpret as we do not know the amount of additional vacuole zinc under those conditions, and the possible changes in utilization mechanism.

6) *Are the low phosphate conditions in Figure 2 truly low phosphate? That is, are the cells growing more slowly than phosphate-replete cells?*

We think so. In the first few generations after the transition, the cells grow about 1.6 times slower than cells growing when Pi is present in the medium.

7) *Figure 4A, C, E. The y-axis is labeled as "Vacuole/cell size" which isn't clear to me; does a value of 1.0 mean the entire cell is vacuole? Given the error bars, it appears that some values were greater than 1.0 so this interpretation cant be correct.*

This interpretation is correct, value of 1 means that the entire cell is vacuole. We now changed the scale and it can be shown that there are only 3 cells that there value was close to one. In these cases the upper side of the error bar is of course limited by this value. We changed the limits accordingly (figure 5d in the revised version)

8) *Figure 5C and D are currently mislabeled "number of cells".*

Corrected

9) *I would add mention of the low copper and low iron effects to the abstract as this is not a zinc-specific effect.*

Added

10) *Figure 2e and f- "phosphate"*

Corrected

11) Figure 2e and f legend. "...for low phosphate conditions..."

Corrected

12) Figure 3a legend. It would help to clarify that the comparison is of mother and daughter cells that are 7 hours old.

This is now clarified in the figure caption

13) Supplemental Figure 1 legend: the descriptions of panels b and c are switched.

Corrected

14) Supplemental Figure 5 legend: FluoZin-1 detects labile zinc only and not "cellular zinc" (the latter suggesting total zinc). Also, the supplemental experimental procedures mentions FuraZin-1 and not FluoZin-1.

Corrected

Reviewer #3 (Remarks to the Author):

*This paper reports the observation that when *Saccharomyces cerevisiae* (budding yeast) cells are exposed to environments containing low concentrations of essential metals (zinc, copper and iron) mother cells continue to produce daughter cells; however, their daughter cells are unable to divide. When the G1/S transition inhibitor *WHI5* is deleted this asymmetry is lost and both mothers and daughters divide in low zinc environments. However, under these conditions *whi5* mother and daughter cells progressively slow in growth resulting in reduced overall population growth rate. The authors show that the growth behavior is correlated with vacuole size and inheritance.*

The regulation of cell growth, including the underlying mechanisms, sources of heterogeneity and evolutionary implications is an important area of research. This paper represents a valuable contribution to the field and capitalizes on the unique capabilities of microfluidic devices and real time imaging of cell growth. The experiments are well executed and the data support the conclusions of the authors. I recommend publication after the authors have addressed the following comments.

1. The authors refer to "limiting zinc conditions". The term "limiting" is not defined and I would expect to see confirmation that final culture density is a function of zinc concentration to justify the claim that the concentration is indeed limiting. I suggest they use "low".

'Limiting' was changed to 'low'.

2. In Figure 1a and 1b it is unclear what the y-axis represents. The authors are measuring microcolonies of cells as shown in Figure 1c. How do they get values in the 100s (Fig 1a) or 10s (Fig 1b) and why are they different? Similarly, in Figure 2 it is unclear if these plots are aggregates of multiple microcolonies or a single microcolony.

This is now better explained in the figure caption. The y axis represents the number of cells within a field of view. In the case of rich conditions the growth is exponential and therefore the number of cells is of the order hundreds. In low zinc they grow linearly reaching lower cell numbers (order tens).

3. I disagree that this "...proliferation pattern represents a general response to nutrient limitation..."

As the authors show this behavior is unique to low concentrations of essential metals and does not occur in phosphate limitation.

The text was modified to emphasize that we see the mother-restricted division pattern in only some stresses and not in others.

4. The authors claim that the daughter cells are "largely devoid of their vacuoles". It seems unlikely (and not proven) that they would have no vacuoles, and more likely that the vacuoles are simply smaller and therefore hard to detect.

"Were largely devoid of their vacuole" was changed to "received very small vacuoles".

5. I find it odd that the authors only introduce the mathematical model in the discussion. Why not include it in the main text?

Model I is now included in main text.

6. The title is misleading as the phenomenon is only observed in a very specific type of stress (and stress is a very subjective classification). Maybe "Budding yeast adopts a mother only proliferation pattern in low metal environments".

Title was changed.

Additional changes

The figures were changed in the following way:

Figure 1 and 2 were reorganized.

Figure 6 was added

Supplementary Figure 2 was moved to the revised manuscript as Figure 7